# Numerical model for noise reduction of small vertical-axis wind turbines

Wen-Yu Wang[1], Yuh-Ming Ferng[2]

[1] Department of Energy and Refrigerating Air-Conditioning Engineering, National Kaohsiung University of Science and Technology, Kaohsiung City, Taiwan

[2] Department of Engineering and System Science, National Tsing Hua University, Hsinchu City, Taiwan

*Correspondence to*: Wen-Yu Wang (wywang@nkust.edu.tw)

Abstract. Small vertical-axis wind turbines are a promising solution for affordable and clean energy, but their noise emissions present a challenge to public acceptance. Numerous blade designs have been aimed at reducing noise but often come with a decrease in wind turbine aerodynamic efficiency. In this study, the acoustic power and torque of a 5 kW vertical-axis wind turbine (VAWT) were simulated by using different mesh sizes and turbulence models. The simulated torque and noise of the turbine have significant sensitivity to the mesh size, so suitable mesh sizes were determined for the near-wall and rotating regions that can be used as a design reference for future turbines with similar operating conditions. The selection of the turbulence model was found to affect the predicted torque by about 10% and the predicted tip noise by about 2 dB. The selected mesh size and turbulence model were then applied to simulating the effectiveness of three common noise mitigation techniques: a mask, deflector, and wall roughness. The results showed that deflectors are suitable for noise reduction of small VAWTs. This paper provides valuable information on simulating noise propagation from small VAWTs and the optimal noise reduction techniques.

## 1. Introduction

Taiwan has poor petrochemical energy resources and relies on imports to meet about 98% of its energy needs. Thus, the active development of green energy has become an increasingly important issue. Owing to Taiwan's geography and monsoon climate, large amounts of wind energy are available. Wind power generation offers significant environmental benefits and is a feasible option as a renewable energy resource. Advances in aerodynamics theory, materials, and manufacturing technologies have improved the performance and reliability of wind turbines while lowering the cost of electricity generation. Around the world, the rise of high-density cities has led to a gradual increase in high-rise buildings, which has increased interest in wind energy for urban environments. A vertical-axis wind turbine (VAWT) offers less power than the conventional horizontal-axis wind turbine (HAWT), but it has a simple design, can operate at low wind speeds, and has low noise emissions. Thus, VAWTs are suitable for urban environments with many buildings. The global installed capacity of small wind turbines in urban areas has been growing annually since 2010 and reached 1427.5 MW in 2020 (Li et al., 2022). However, the spread of wind power generation is limited not only by the terrain but also by the noise generated by the wind and turbine blades, which may affect nearby residents.

The analysis of aero-acoustic noise is a mature research topic. Lighthill (1952) derived an acoustic wave equation by using fluid mechanics theory, where the sound source term can be obtained experimentally or by computational fluid dynamics (CFD). Proudman et al. (1952) improved upon Lighthill's work and derived the sound source caused by isotropic turbulence. Pradera et al. (2007) calculated various aerodynamic parameters and noise generated by viscous fluids with low and high Reynolds numbers passing over a cylinder and used the Ffowcs Williams–Hawkings (FW–H) equation to predict the sound pressure, which they then transformed into the sound pressure level (SPL) through the fast Fourier transform for noise analysis. SPL is the most commonly used indicator of the acoustic wave strength. It is a logarithmic measure of the effective pressure of a sound relative to a reference value, defined in dB. Their analytical results showed good agreement with the experimental data.

Various experimental and numerical techniques have been developed for mitigating the noise emissions of wind turbines based on their aero-acoustic characteristics. Some promising noise mitigation techniques targeting dominant noise sources have been discussed, including reducing the inflow turbulence noise, trailing edge noise, and tip noise. Although the noise produced by VAWT is lesser than HAWT, the potential noise pollution should not be neglected. Generally, the noise generated by operational wind turbines can be categorized into mechanical noise and aerodynamic noise. Mechanical noise is generated by various machinery components, such as caterpillar bands, gearbox, and generator. On the other hand, aerodynamic noise is generated from the moveing blades and is primarily associated with the interaction of turbulence with the blade surface. An H-VAWT would have very different tip noise to a Darrieus troposkein. Mechanical noise can be effectively reduced by engineering methods, while aerodynamic noise remains a challenging issue (Ghasemian et al., 2017). Maizi et al. (2018) performed a 3D numerical analysis with unsteady CFD simulations (URANS and DES) showing that a shark tip reduced the tip noise by 7% compared with the reference tip but with a tradeoff of 3% less power. However, their computational approach involved using a detached eddy simulation to resolve the flow field and the FW–H equation for acoustic calculations, which was very computationally intensive. Deshmukh et al. (2018) included the tip region in an annular domain for a parametric study of blended winglets to evaluate the improvement in the aerodynamic and aero-acoustic performances. Their methodology significantly reduced the computational cost, and their results showed that noise emissions were reduced by about 25% at mid-high frequencies along with enhanced torque output. Mohamed (2016) used a CFD model combined with an unsteady realizable $k$–$\varepsilon$ turbulence model to analyze the noise and efficiency of H-rotor Darrieus VAWT with different azimuthal spacings between the airfoils in every blade at different tip speed ratio. The results showed that, compared with a single blade, a double blade with 60% spacing effectively reduced the noise by 40% across the entire frequency range, but the efficiency and torque were decreased. Botha et al. (2017) compared the noise emissions of a six-bladed 2 kW helical VAWT measured in experiments with the two-dimensional analytical solution and the CFD predictions. They solved RANS and DES equations in 2D and 3D simulations on ANSYS FLUENT. Their calculations demonstrate that ANSYS FLUENT give accurate noise projections compare to analytical models. They suggest that the inflow turbulence noise can be regarded as the main noise source. Naccache et al. (2017) performed 2D experiments to verify their CFD model. The results showed that using a shear stress transport SST $k$–$\omega$ turbulence model with a near-wall mesh of $y^+ < 15$ could accurately predict the lift coefficient, lift–drag ratio, and power coefficient at different azimuth angles and rotational speed ratios. This model was then applied to conducting in-depth 3D simulations. Manuel et al. (2020) performed a 3D Large Eddy Simulations (LES) and aeroacoustic spectra for three selected configurations: an isolated NACA0012 airfoil, an isolated rotating VAWT and a farm of four VAWT were simulated. This study has aided in pointing the sources of noise in different setups and associating them to the physical mechanisms responsible for aeroacoustic generation in VAWTs and arrays of turbines. Weber et al. (2015) validate two different numerical methods for noise prediction of the Darrieus turbine with 3 blades of NACA0018 cross section with a chord length of 0.05 m using a complementary approach consisting of experimental measurements and numerical simulations. Venkatraman et al. (2021) performed a 2D URANS numerical investigation of the effect of inflow on the noise radiated by a VAWT compared with the experimental data presented in Weber et al. (2015) based the CFD software CFX 19.1 with SST $k$–$\omega$ turbulence model. Excellent agreement was found for the first two Blade Passing Frequencies (BPF) with good agreement for the next BPF and the broadband noise level. They were using two different numerical schemes for noise prediction using hybrid methods. Both methodologies were compared with experimental data.

Previous studies numerically investigated the low-frequency emissions of a generic 5 MW wind turbine and evaluated the influence of a tower and steady blade deformation under uniform inflow conditions. Klein et al. (2018) coupled the CFD solver FLOWer to the multibody simulation (MBS) solver SIMPACK with 3D RANS solver, which they applied to minimizing the noise emissions of a wind farm by changing the operating modes of individual wind turbines. Abreu et al. (2022) used advanced 3D numerical techniques to study whether the ground structure on the wave path between a wind turbine and seismic station can be changed to reduce or mitigate the noise emissions of the wind turbine. They showed that filling trenches with water and

relatively simple changes to the topography helped reduce noise emissions. Chen et al. (2021) designed two types of deflectors to enhance the performance of a three-bladed VAWT and found that the optimized upper deflector improved the performance by 20% and the optimized lower deflector improved the performance by 17%.

The objective of the present study was to analyze the noise emissions of a 5 kW VAWT and the effects of different noise reduction techniques on not only the noise emissions but also the aerodynamic torque and acoustic power. The effects of different steady-state turbulence models and the mesh size on the results were evaluated, and the optimal mesh size and model were then applied to analyzing three different noise reduction techniques. This is a qualitative study looking at changing VAWT design to improve performance and reduce noise. The accurate environmental noise impact of a VAWT is not discussed in this study.

## 2. Methods

### 2.1 Numerical method

Blade design for small VAWTs must consider both the power generated and noise emitted. The CFD code, ANSYS FLUENT, is commercially available and an industrial leading software used to simulate the aerodynamic performance of wind turbine airfoils and aeroacoustic analysis. (Yao et al., 2012, Zaareer et al., 2023) In this study, the aerodynamic flow parameters required on and around the blade surfaces for the FW–H codes were obtained using 3D URANS based CFD solver ANSYS FLUENT. It was used to simulate turbulence and acoustic models to analyze the causes of noise emissions. The governing equations were as follows. The flow velocity was much less than the speed of sound, so the aerodynamic flow field can be considered incompressible. Therefore, the continuity equation can be written as

$$\frac{\partial \rho}{\partial t} + \nabla \cdot (\rho u) = 0 \tag{1}$$

where $\rho$ is the density and $u$ is the velocity. The momentum equation is written as

$$\rho(\frac{\partial u}{\partial t} + u \cdot \nabla u) = -\nabla p + \mu \nabla^2 u + \rho g \tag{2}$$

Where t is time, $p$ is the static pressure, $\mu$ is dynamic viscosity, and $\rho g$ is the body force.

### 2.2 Turbulence models

Based on the review of relevant research (Venkatraman et al. (2021); Mohamed (2016)), A URANS model is achieved with the realizable $k$–$\varepsilon$ model and SST $k$–$\omega$ turbulence models. The flow solution is then coupled to an acoustic solver, based on the FW-H analogy for the prediction of noise. These models modify their original two-equation versions to address phenomena such as vortices, wake flows, and flow separation. These models can be used to simulate the turbulence generated by a blade passing through the wind field reliably and economically. The realizable $k$–$\varepsilon$ model is considered more accurate than the standard $k$–$\varepsilon$ model at predicting the dissipation rate distribution and boundary layer characteristics of separated and recirculating flows. The turbulence kinetics is expressed as

$$\frac{\partial}{\partial t}(\rho k) + \frac{\partial}{\partial x_j}(\rho k u_j) = \frac{\partial}{\partial x_j}\left[\left(\mu + \frac{\mu_t}{\sigma_k}\right)\frac{\partial k}{\partial x_j}\right] + G_k + G_b - \rho \epsilon - Y_M + S_k \tag{3}$$

The dissipation rate is expressed as

$$\frac{\partial}{\partial t}(\rho \epsilon) + \frac{\partial}{\partial x_j}(\rho \epsilon u_j) = \frac{\partial}{\partial x_j}\left[\left(\mu + \frac{\mu_t}{\sigma_\epsilon}\right)\frac{\partial \epsilon}{\partial x_j}\right] + \rho C_1 S \epsilon - \rho C_2 \frac{\epsilon^2}{k + \sqrt{v\epsilon}} + C_{1\epsilon}\frac{\epsilon}{k}C_{3\epsilon}G_b + S_\epsilon \tag{4}$$

where $G_k$ represents the generation of turbulence kinetic energy due to the mean velocity gradients; $G_b$ represents the generation of turbulence kinetic energy due to buoyancy; $G_\omega$ represents the generation of $\omega$; $Y_M$ represents the contribution of the fluctuating dilatation in compressible turbulence to the overall dissipation rate; $C_{1\varepsilon}$ and $C_2$ represent constants; $\sigma_k$ and $\sigma_\varepsilon$ represent the turbulent Prandtl numbers for k and ε, respectively. The model constants are $C_{1\varepsilon}=1.44$, $C_2=1.9$, $\sigma_k=1.0$, and $\sigma_\varepsilon=1.2$. $S_k$ and $S_\varepsilon$ represent user-defined source terms. All constants are given in the Table 1.

The SST $k–\omega$ model works well in areas near and far from the wall, and it can be used at low and high Reynolds numbers. It is more nonlinear than the $k–\varepsilon$ model and has more difficulty in converging. The model provides a better prediction of flow separation than most RANS models, which accounts for its good performance with adverse pressure gradients and is why it is frequently applied in aerodynamics. The turbulence kinetics energy is expressed as (Menter,1994)

$$\frac{\partial k}{\partial t}+U_j\frac{\partial k}{\partial x_j}=P_k-\beta^* k\omega+\frac{\partial}{\partial x_j}\left[\left(v+\sigma_k v_T\right)\frac{\partial k}{\partial x_j}\right]$$

(5)

The specific dissipation rate is expressed as

$$\frac{\partial \omega}{\partial t}+U_j\frac{\partial \omega}{\partial x_j}=\alpha S^2-\beta\omega^2+\frac{\partial}{\partial x_j}\left[\left(v+\sigma_\omega v_T\right)\frac{\partial \omega}{\partial x_j}\right]+2\left(1-F_1\right)\sigma_{\omega 2}\frac{1}{\omega}\frac{\partial k}{\partial x_i}\frac{\partial \omega}{\partial x_i}$$

(6)

where

$$v_T=\frac{k}{\omega}$$

(7)

$$F_1=\tanh\left\{\left\{\min\left[\max\left(\frac{\sqrt{k}}{\beta^*\omega y},\frac{500v}{y^2\omega}\right),\frac{4\sigma_{\omega 2}k}{CD_{k\omega}y^2}\right]\right\}^4\right\}$$

(8)

$$P_k=\min\left(\tau_{ij}\frac{\partial U_i}{\partial x_j},10\beta^* k\omega\right)$$

(9)

$\alpha_1$, $\alpha_2$, $\beta_1$, $\beta_2$, $\beta^*$, $\sigma_{k1}$, $\sigma_{k2}$, $\sigma_{\omega 1}$, and $\sigma_{\omega 2}$ represent constants and were given in the Table 1.

**Table 1. Constant parameters for realizable $k–\varepsilon$ and SST $k–\omega$ turbulence models**

| | Symbol | Default Value |
|---|---|---|
| **realizable $k–\varepsilon$** | $C_{1\varepsilon}$ | 1.44 |
| | $C_2$ | 1.9 |
| | $\sigma_k$ | 1.0 |
| | $C_{1\varepsilon}$ | 1.44 |

| | | |
|---|---|---|
| | $\sigma_\varepsilon$ | 1.2 |
| **SST $k$–$\omega$** | $\alpha_1$ | 5/9 |
| | $\alpha_2$ | 0.44 |
| | $\beta_1$ | 3/40 |
| | $\beta_2$ | 0.0828 |
| | $\beta^*$ | 0.09 |
| | $\sigma_{k1}$ | 0.85 |
| | $\sigma_{k2}$ | 1 |
| | $\sigma_{\omega1}$ | 0.5 |
| | $\sigma_{\omega2}$ | 0.856 |

### 2.3 Aero-acoustic formulation

Proudman (1952) applied Lighthill's acoustic theory to complete flow fields for the first time to derive the noise generated from isotropic turbulence with low Mach numbers ($M_t$) and high Reynolds numbers. However, the noise source generated by turbulent flow can be described as: (Lilley,1994)

$$p_A = \alpha_\varepsilon \rho_0 \varepsilon M_t^5 \tag{10}$$

Where $p_A$, $\alpha_\varepsilon$, $\rho_0$, and $\varepsilon$ represent acoustic power, density of the far field, an empirical constant (=0.1), and mean rate of dissipation of energy per mass, respectively. The turbulent $M_t$ can be expressed as:

$$M_t = \frac{\sqrt{2k}}{\alpha_0} \tag{11}$$

Where $\alpha_0$ is the sound speed and $k$ is the turbulence kinetic energy. It should be noted that this sound source
model assumes isotropic turbulence and only considers the energy of turbulent disturbances. Although it cannot provide sound sources on the frequency spectrum, it is sufficient to evaluate the magnitude of the noise and the results of subsequent noise reduction. On the other hand, the turbulent boundary layer generated by the motion of an object will also produce noise sources on the object surface due to its disturbances. In order to reduce noise produced during small VAWT operation, CFD analysis for different wind turbine blades'
attack angles, coupled with the noise analysis is performed. Sound propagation equation is solved by Lighthill and Curle (Dinulovic,2023). FW-H and Curle's analogy are the most widely used integral methods for predicting acoustic field and considered as formal solutions of Lighthill's equation applied for rigid and moving wall boundaries relative to flow field (Nukala et al., 2023). The methodology we used for aero-acoustics has been previously applied to VAWTs. The magnitude can be described as: (Curle,1955)

$$P_A = \int_S I \, dS \tag{12}$$

Where $P_A$, $I$, and $S$ are the acoustic power, sound intensity, and control surface on the moving object, respectively. $I$ can be expressed as:

$$I = \frac{A_c}{12\rho_0 \pi a_0^3} \overline{\left(\frac{\partial p}{\partial t}\right)^2} \tag{13}$$

where $A_c$ is the correlation area. It represents the region within which the noise patterns maintain a certain level of similarity or correlation before dissipating or changing significantly due to various factors like environmental conditions. $I$ can be interpreted as the local contribution per unit surface area of the body surface to the total acoustic power. The mean-square time derivative of the surface pressure and the correlation area are further approximated in terms of turbulent quantities. Generally, the noise generated from the blade surface is the main source of noise, while the noise caused by turbulence is relatively small. Therefore, this study mainly focuses on the noise generated by the blade.

### 2.4 CFD model

The Darrieus H-VAWT considered in this study had three blades, each with a S4415 airfoil shape, chord length of 0.7 m, height of 5.5 m, and rotation radius of 2.235 m, as shown in Fig. 1(a). For the CFD model, all of the blades were included in the computational domain, and the blade surfaces were set as stationary walls for the boundary condition. To simulate the rotation of a VAWT, the computational domain was divided into a rotating region and outer-flow region. As shown in Fig. 1(b), the rotating region was a dynamic mesh that rotated at a frequency of 60 rpm in the shape of a vertical cylinder with a radius of 3 m and height of 6.5 m, and it included the three blades. Dynamic mesh allows for adaptability to complex geometries without requiring re-meshing. The mesh is adapted dynamically during the simulation to capture changes in the flow domain. An implicit scheme is employed to calculate the transient terms, so the acoustic cases are less sensitive to the CFL number. The interface between the rotating region and outer-flow region was set to the interface to ensure the continuity of the velocity and pressure between them. The geometric shape of the outer-flow region should have little influence on the results because it is almost unaffected by the blades. Figure 1(c) shows the outer-flow region; to simplify the mesh generation process, it was set as a horizontal cylinder with a length of 40 m and radius of 5 m, where the top surface was set as the velocity inlet with a standard wind speed of 12 m/s, the bottom surface was set as the pressure outlet with a pressure of 0 Pa, and the side surfaces were set to the symmetric boundary condition. Cylindrical domains are often employed for simulating individual wind turbine blades or small wind turbine systems. The geometric similarity between cylinders and wind turbine blades allows for a better representation of the airflow around the blades. However, rectangular domains are more suitable for simulating larger wind fields, such as interactions between multiple wind turbines or airflow distributions within a wind farm.

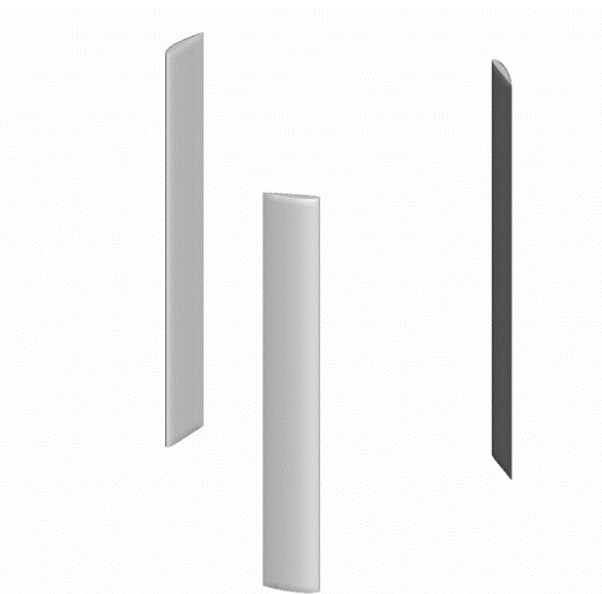

(a)

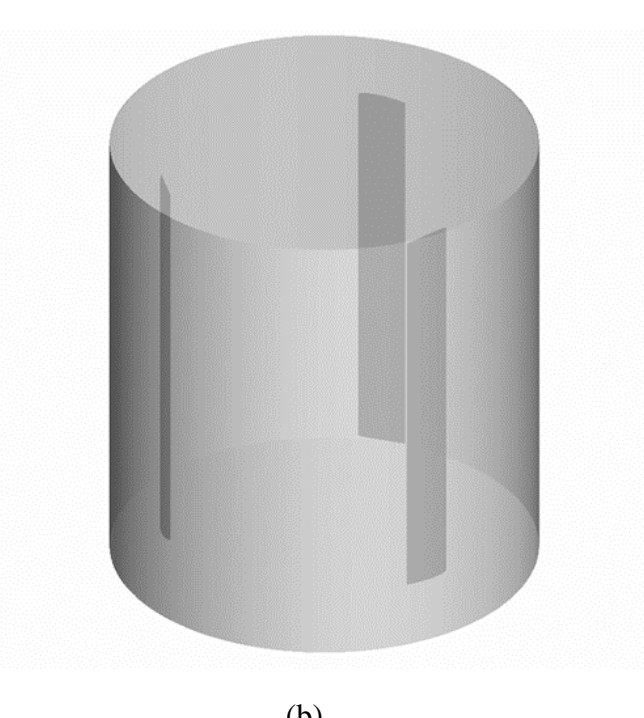

(b)

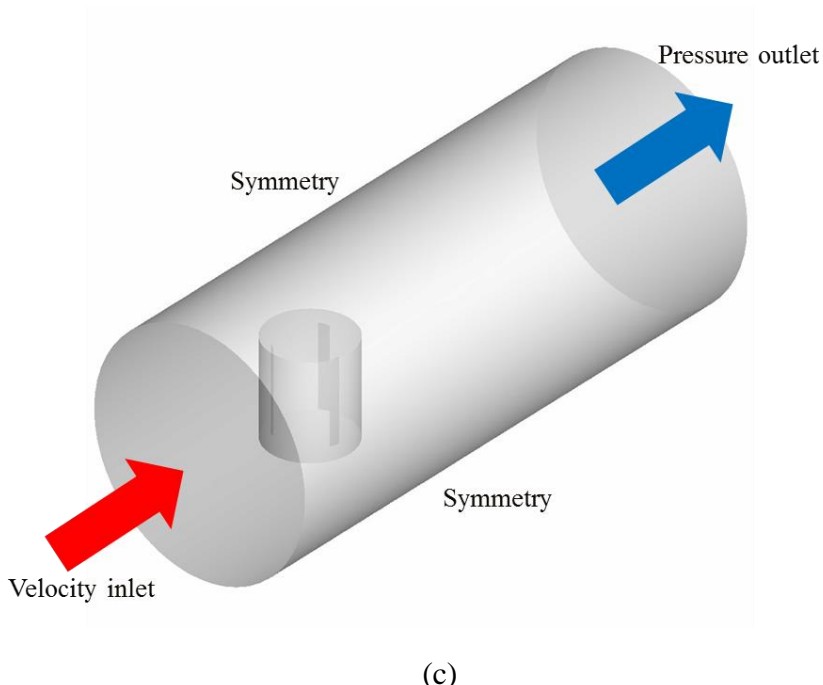

Pressure outlet

Symmetry

Symmetry

Velocity inlet

(c)

**Figure 1: CFD model: (a) blade geometry, (b) rotating region, and (c) outer-flow region.**

## 2.5 Mesh division

The momentum equation has a significant impact on the wind turbine torque. Thus, both the pressure and velocity were discretized by using the second-order upwind scheme, and the SIMPLE algorithm was used to

couple the pressure and velocity. The turbulent kinetic energy and dissipation were discretized by using the first-order upwind scheme. First-order schemes may be computationally efficient and the continuity equation is easy to converge. Their accuracy for simulating turbulence depends on the specific application and the turbulence model. The convergence of turbulent flow in the cases presented in this paper does not exhibit significant problem with turbulent flow convergence. The time was discretized by using an implicit first-order differential equation with a time step of 0.01 s. An implicit scheme is employed to calculate the transient terms, so the acoustic cases were less sensitive to the CFL number in this paper. CFL number is acceptable less than 200 based on the FLUENT manual. Therefore, the CFL numbers 0.12 to 2 and a time step of 0.01 s were used in the paper. Spatial discretization is more complex because the flow field has different characteristic lengths inside and outside the boundary layer. Therefore, the entire domain was divided into three regions, all with different mesh sizes: the near-wall region, rotating region, and outer-flow region. The near-wall region was the first mesh layer on the blade surface, and the mesh needed to be very dense to facilitate the simulation of the velocity gradient and shear stress on the wall surface. SST k–ω model has a higher tolerance for y+, and realizable k–ε has s maximum y+ value less than 80 in this paper. The rotating region also required a dense mesh to capture the wake and vortex formed after the airflow passed around the blade. The outer-flow region was not affected by the blades, nor did it affect the blades. Thus, a coarser and unstructured tetrahedral mesh could be used here. As shown in Fig. 2, the mesh sizes in the near-wall and rotating regions were varied to evaluate the effect on the simulation results. The mesh type was not changed. An unstructured mesh was used in the outer-flow region (Fig. 2(a)), and a higher-quality structured mesh was used in the more complex rotating region (Figs. 2(a)–(d)). Table 2 presents the characteristic lengths of the four mesh sizes in the near-wall and rotating regions. We performed a mesh independence study based on the ASME standard (ASME V&V 20-2009) in this study, and the characteristic length of the meshes were varied by about 1.3 times to evaluate the significance of different mesh sizes.

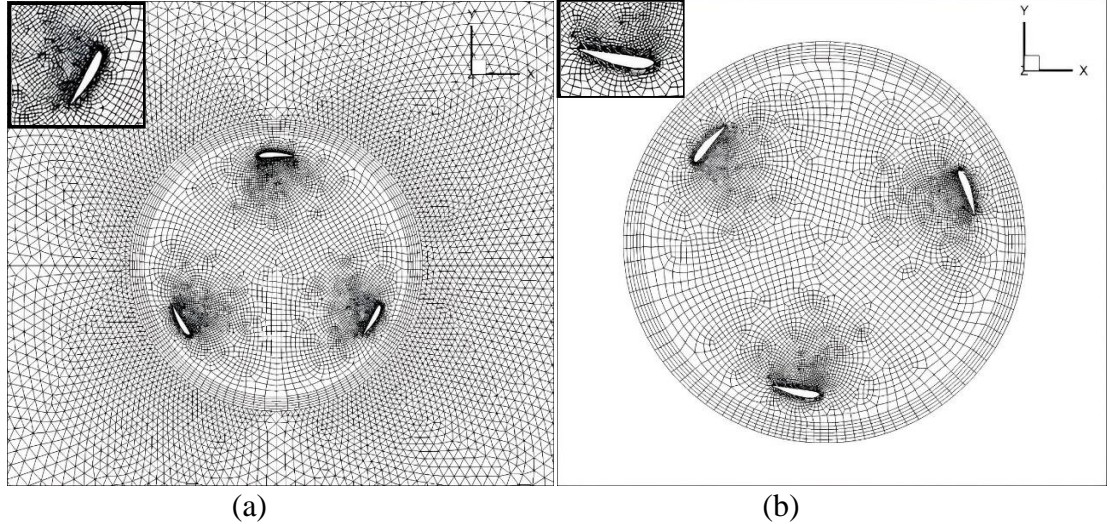

|     |     |
| --- | --- |
| (a) | (b) |

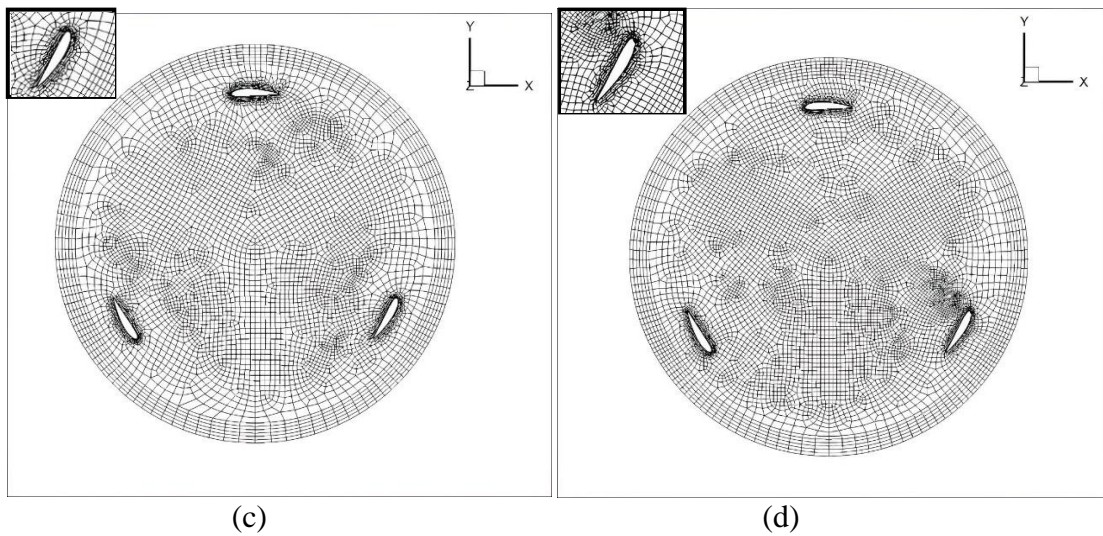

| | (c) | (d) |

**Figure 2: Divisions of (a) Mesh 1, (b) Mesh 2, (c) Mesh 3, and (d) Mesh 4.**

**Table 2. Characteristic lengths of different meshes in each region.**

| | Near-wall region (m) | Rotating region (m) | Outer-flow region (m) |
|---|---|---|---|
| Mesh 1 | 0.01 | 0.1 | 1 |
| Mesh 2 | 0.003 | 0.1 | 1 |
| Mesh 3 | 0.003 | 0.08 | 1 |
| Mesh 4 | 0.003 | 0.06 | 1 |

## 3. Results and Discussion

CFD was used to simulate the torque and acoustic power distributions of a small VAWT in a standard wind speed 12 m/s and rotational speed 60 rpm in this study. The effects of the mesh and turbulence models were analyzed, and the best combination was applied to analyzing the flow field and identifying mechanisms for torque reduction and increased noise. Finally, three commonly used noise reduction techniques were evaluated for their effectiveness.

### 3.1 Effects of the mesh and turbulence models

Figure 3(a) shows the results of the mesh independence test. T is the time for one rotation. Different t/T values correspond to blade positions. 60 distinct revolutions of the turbine were simulated and took approximately 1 minute for the initial transient. The predicted torque was quite sensitive to the mesh size, which could cause differences of over 50%. Torque is computed by multiplying the aerodynamic force on the blades by the distance from the center of rotation to the point where the force acts. Fortunately, mesh 270 independence could still be achieved by adjusting the mesh size in different regions, and mesh independence was achieved with Mesh 3 and Mesh 4. Similar results were obtained for the predicted acoustic power on the blade, as shown in Fig. 3(b). Mesh 3 and Mesh 4 resulted in nearly identical predictions, so Mesh 3 was

selected for subsequent analysis because it had a smaller grid number and thus was less computationally intensive. The relationship between torque and acoustic power trends in a system can be complex and is influenced by various factors related to the mechanical and fluid dynamics of the system. CFD simulations may not capture all relevant frequency components, particularly those high-frequency acoustic phenomena. Insufficient frequency resolution can lead to an incomplete representation of the acoustic spectrum, affecting the relation with torque. Thus, the acoustic power and torque plots don't have similar periods. Each solution procedure was found to take approximately within 10 h of central processing unit (CPU) time when executed on an Intel i9-13900k high-performance workstation to achieve a converged solution. Figure 3(c) shows that the torque predicted by the realizable $k$–$\varepsilon$ and SST $k$–$\omega$ turbulence models were very similar. Both models obtained three peaks and valleys in one cycle, and the corresponding time points were almost identical except that the maximum value predicted by the realizable $k$–$\varepsilon$ turbulence model was 11% higher than that predicted by the SST $k$–$\omega$ turbulence model. In addition, the realizable $k$–$\varepsilon$ turbulence model predicted a time-averaged torque of 227.7 N·m, which was slightly higher than the value of 207.2 N·m predicted by the SST $k$–$\omega$ turbulence model. However, the two turbulence models showed significant differences in the predicted acoustic power, as shown in Figure 3(d) The time-averaged energy predicted by the realizable $k$–$\varepsilon$ turbulence model was 57% higher than that predicted by the SST $k$–$\omega$ turbulence model. However, from an acoustic point of view, the difference between the two predictions was about 1.5 W, which is still acceptable. Previous studies have shown that the SST $k$–$\omega$ turbulence model has a higher tolerance for $y+$ and is more suitable for the geometry of different noise reduction techniques. (Menter, 1994; Menter, 2012) Therefore, the SST $k$–$\omega$ turbulence model was selected for subsequent analyses in this study.

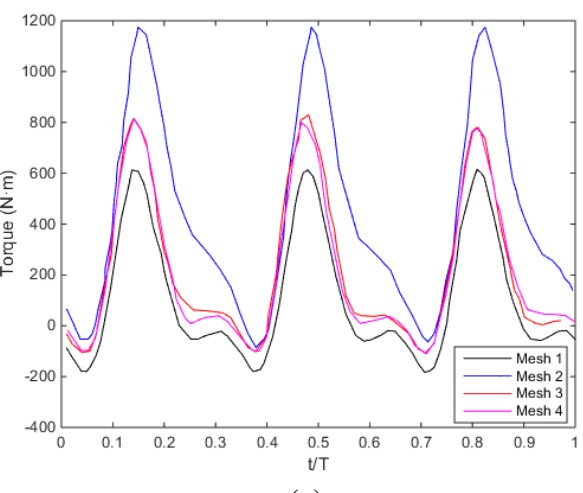

(a)

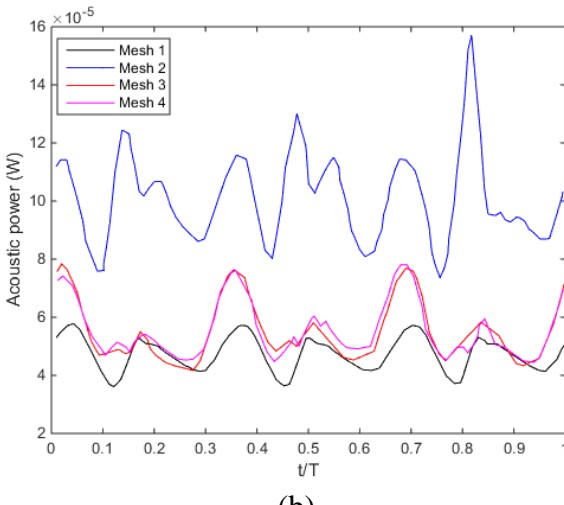

(b)

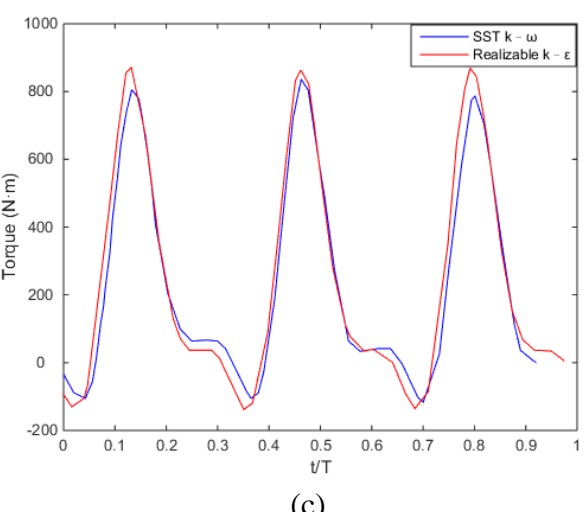

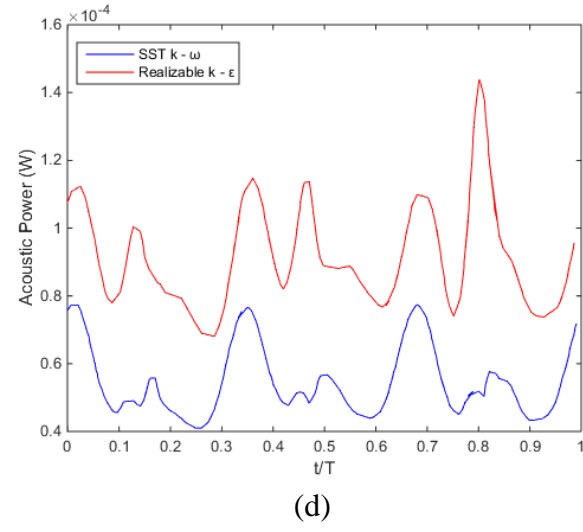

(c)

(d)

**Figure 3: Effects of the mesh on the predicted (a) torque and (b) acoustic power. Effects of the turbulence model on the predicted (c) torque and (d) acoustic power.**

### 3.2 Simulation without noise reduction

Figure 4(a) shows the predicted torque and noise for one cycle without noise reduction. The time-averaged torque and noise sources on the blade surface were 207.2 N·m and $5.8 \times 10^{-5}$ W, respectively. These values were used as benchmarks for evaluating the noise reduction techniques. The maximum noise coincided with the minimum torque while the maximum torque increased noise to a much lesser extent than the minimum torque. This phenomenon was further analyzed by the pressure and streamline distributions on the cross-section combined with the turbulent kinetic energy distribution to further understand the characteristics of the flow field. Figures 4(b) and (c) show the pressure distributions and streamlines at $t/T = 0.70$ and 0.82, respectively. The wind flowed from left to right, and the blade rotated counterclockwise. At $t/T = 0.70$, the attack angles of the three blades were $-132°$, $-12°$, and $108°$. The attack angle is the angle at which the chord of a blade meets the wind velocity. The wind velocity was sampled at a point. When air passed over the blade with an attack angle of $-132°$, a vortex was generated behind the inner edge, which caused a low pressure zone that made it difficult for the blade to rotate. Similarly, when the air passed over the blades with attack angles of $-12°$ and $108°$, high pressure zones were generated in front that made it difficult for the blades to rotate. This resulted in a negative torque, which means that an external force was needed to maintain the rotational speed. Because of the symmetry of the three blades, this phenomenon also occurred at $t/T = 0.04$ and 0.37. At $t/T = 0.82$, the attack angles of the blades were $-55.2°$, $-175°$, and $64.8°$. For the blade with an attack angle of $-55.2°$, the airflow generated a relatively high pressure on the outside and a large low pressure zone on the inner leading edge, which drove the blade to produce a positive torque. No significant pressure differences were observed for the blades with attack angles of $-175°$ and $64.8°$. The noise source caused by turbulence was proportional to the fifth power of the turbulent kinetic energy. Therefore, the turbulent kinetic energy distribution could be used to further analyze the noise caused by the flow field, as shown in Fig. 4(d). At $t/T = 0.70$, the vortex generated on the inner edge of the blade with the $-132°$ attack angle not only reduced the torque of the wind turbine but also generated a large amount of turbulent kinetic energy, which produced the largest noise source. The opposing distributions of torque and noise suggest that the VAWT can be designed to enhance torque and reduce noise simultaneously.

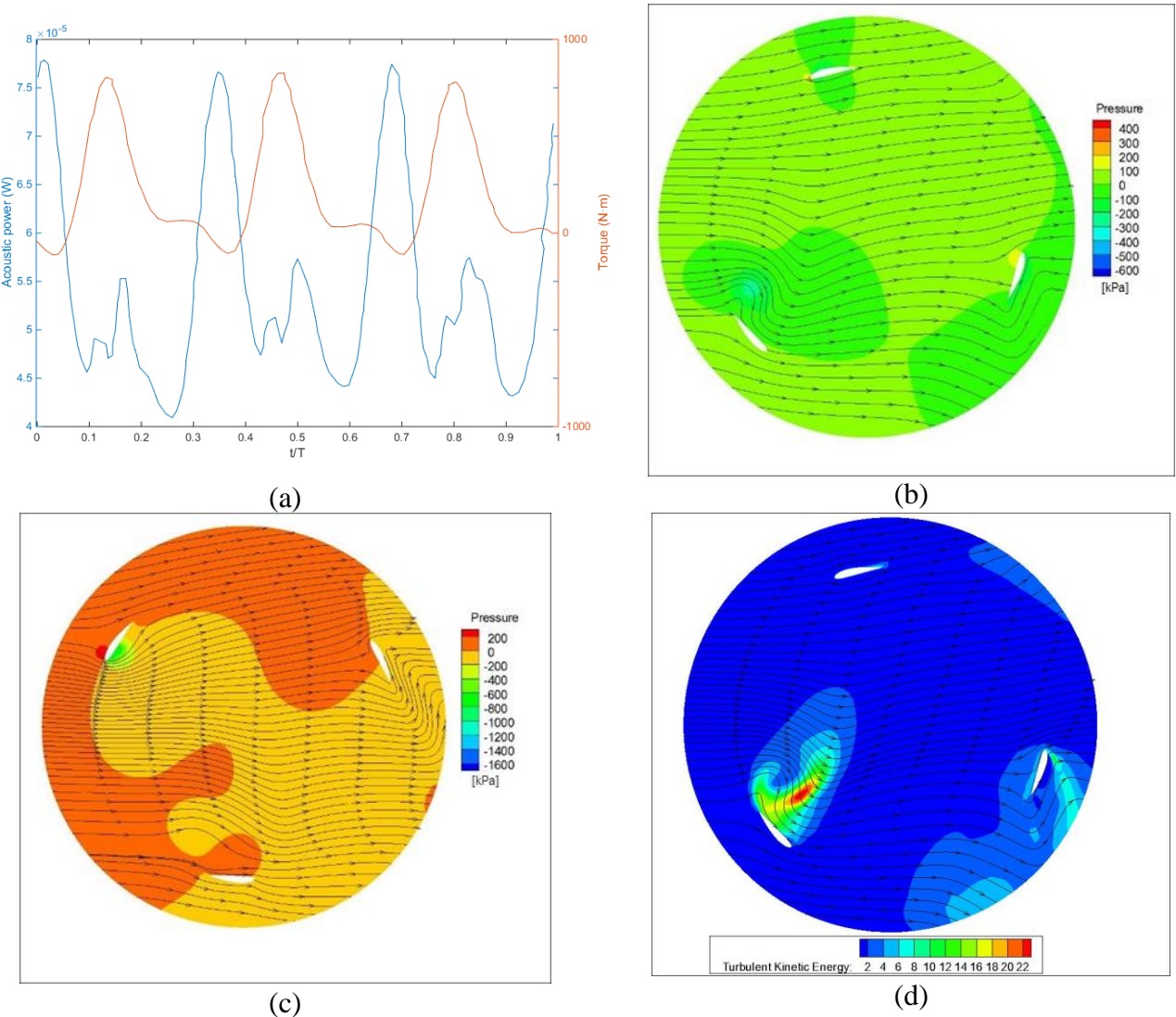

(a)

(b)

(c)

(d)

**Figure 4: Simulation results without noise reduction: (a) comparison between the acoustic power and torque, pressure distributions and streamlines at (b) *t/T* = 0.70 and (c) t/T = 0.82, and (d) turbulent kinetic energy distribution at *t/T* = 0.7.**

### 3.3 Simulation with noise reduction

Three commonly used noise reduction techniques were considered: a mask, deflector, and wall roughness. Mesh 3 and the SST $k$–$\omega$ turbulence model were used under the same wind speed and rotational speed conditions as for the simulation without noise reduction. The simulated torque and blade acoustic power were compared with the benchmark values to evaluate the effectiveness of each noise reduction technique.

### 3.3.1 Mask

Installing a mask on the upper and lower ends of a blade surface prevents sharp angles that cause strong separation or vortices. In this study, a mask with a long axis of 1.2 m and short axis of 0.5 m was installed at the upper and lower ends of the blades, as shown in Fig. 5. Figure 6(a) shows that installing the mask decreased the average torque by 40% to 124. However, Fig. 6(b) shows that the mask did not result in a corresponding decrease in the noise. The average acoustic power after installation was $6.34 \times 10^{-5}$ W, which

is a 16% increase. As discussed in Sect. 3.2, the noise source was the vortex generated inside the blade after the blade cut through the wind field. Therefore, adding a mask at both ends not only failed to reduce noise but also increased the area of friction with the air, which actually increased the noise and resistance.

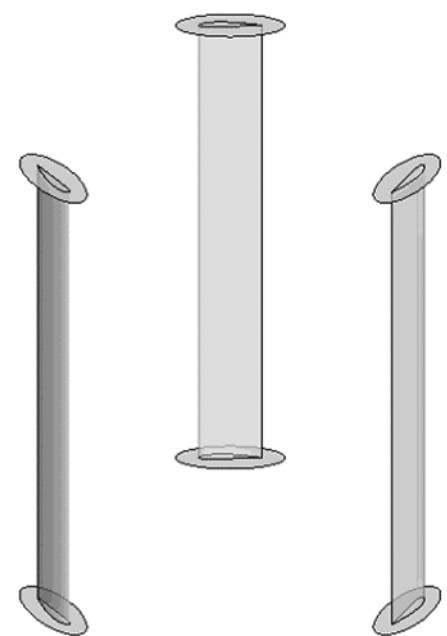

**Figure 5: Geometric design of the mask.**

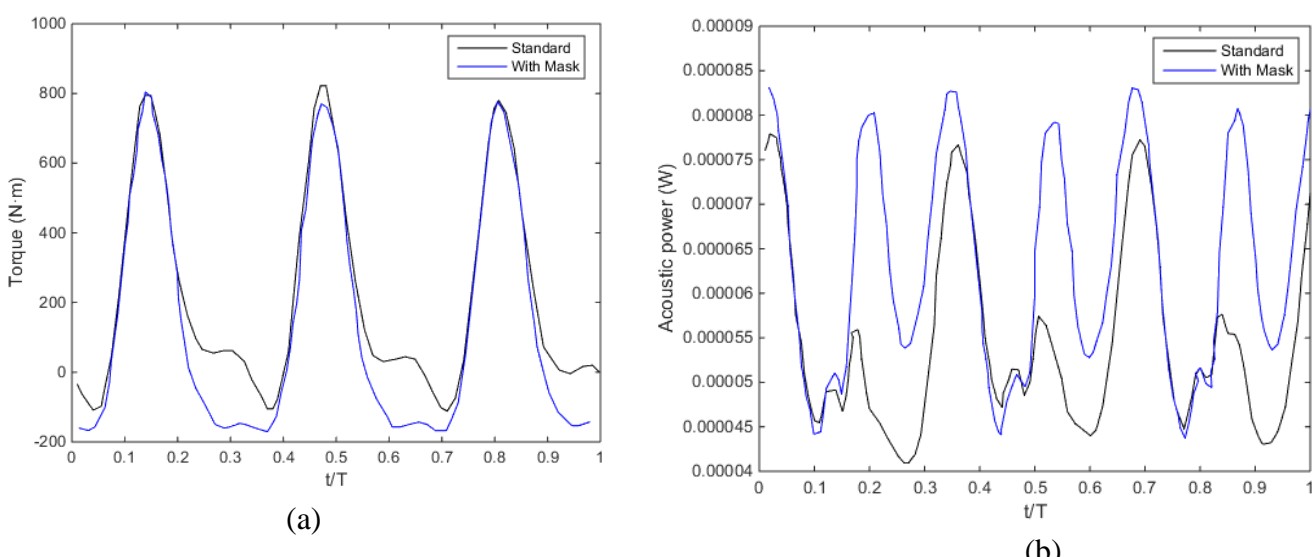

(a)

(b)

**Figure 6: Effects of a mask on the (a) torque and (b) acoustic power.**

### 3.3.2 Deflector

Figure 7 shows the deflector, which is a rectangular baffle with a length of 0.1 m, width of 0.2 m, and spacing of 0.2 m. Figures 8(a) and (b) show the effects of installing the deflector on the torque and blade noise, respectively. Installing the deflector increased the average torque by 169% to 564 and decreased the blade noise by 98% to $7.86 \times 10^{-7}$ W. Figure 8(c) shows the streamlines and pressure distribution after the deflector was installed when the torque was at its minimum (i.e., $t/T = 0.7$). Compared to the pressure

distribution without the deflector (Fig. 4(b)), the vortex at the inner edge of the blade with an attack angle of $-132°$ detached further away from the blade surface, which resulted in a less significant impact on the blade surface and enhanced the torque. Figure 8(d) shows the turbulence kinetic energy distribution at this time and clearly indicates that the vortex separated from the blade. In contrast, the turbulence distribution without the deflector (Fig. 4(c)) shows that the vortex adhered closely to the inside of the blade and continued to affect the blade surface. However, installing the deflector did generate greater turbulent kinetic energy throughout the rotation area, which increased the turbulent noise and thus needed to be evaluated. Figure 8(e) shows the noise energy caused by turbulence in the rotating region. Installing the deflector increased the average turbulent noise kinetic energy about 10.6 times to $2.95 \times 10^{-6}$. After the installation of the deflector, the main noise-generating mechanism changed from boundary layer disturbance to turbulence generated noise. However, the noise generated by turbulence after deflector installation was still an order of magnitude less than the blade noise before installation. This indicates that installing a deflector would be effective for noise reduction.

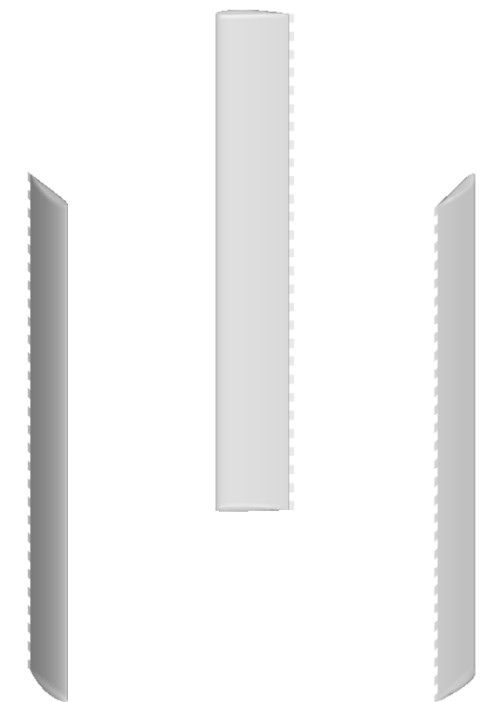

**Figure 7: Geometric design of the deflector.**

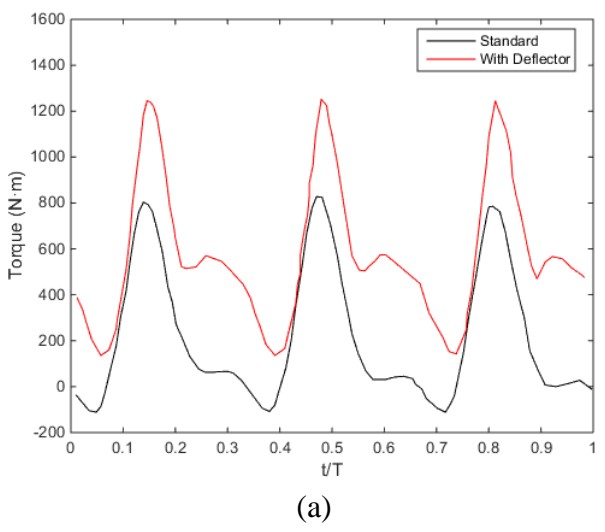

(a)

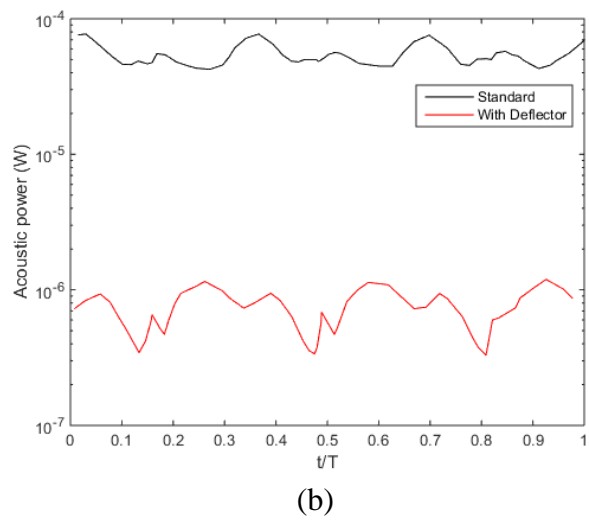

(b)

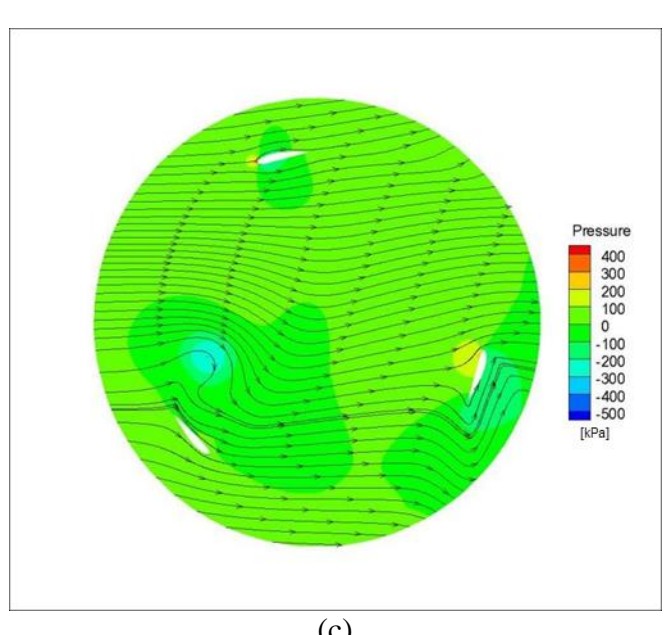

(c)

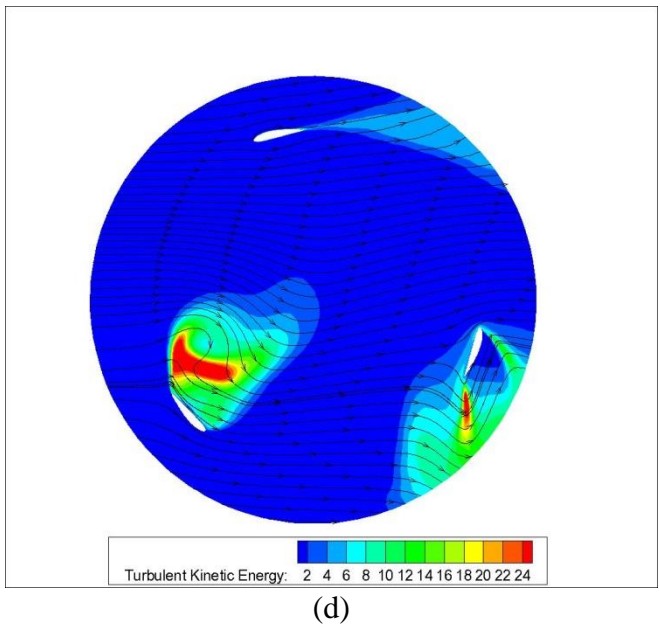

(d)

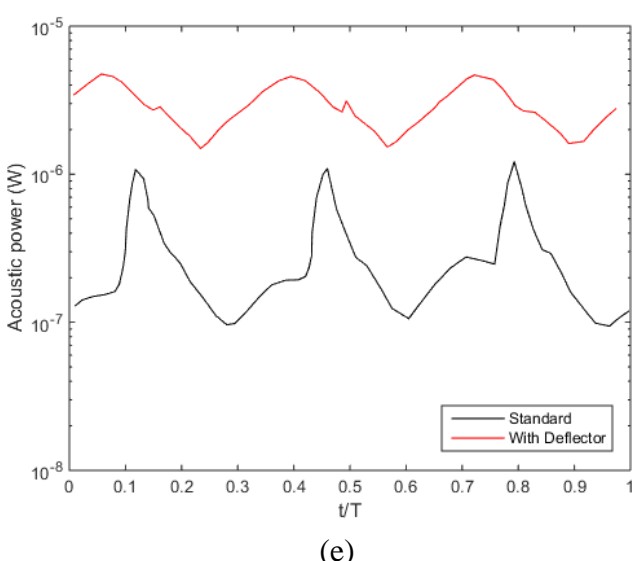

(e)

**Figure 8: Effects of a deflector on the (a) torque, (b) blade surface noise, (c) pressure distribution and streamlines at *t/T* = 0.7, (d) turbulent kinetic energy distribution at *t/T* = 0.7, and (e) turbulent flow noise.**

### 3.3.3 Wall roughness

To achieve noise reduction, the blade surface can be roughened to be similar to that a golf ball. This suppresses the boundary layer separation and reduces the vortex scale, which reduces noise emissions. In this study, the boundary conditions for the rough blade surface were described by modifying the wall function to account for greater wall shear stress on rough surfaces:

$$u^+ = \frac{1}{\kappa}\log y^+ - \Delta B \tag{14}$$

where $\kappa$ and $\Delta B$ are the Von Karman constant and roughness function. With different roughnesses, $\Delta B$ can be expressed as

$$\Delta B = \begin{cases} 0 & \text{for } K_s^+ \leq 2.25 \\ \frac{1}{\kappa}\ln\left(\frac{K_s^+ - 2.25}{87.75} + C_s K_s^+\right) \times \sin\left[0.4258\left(\ln K_s^+ - 0.811\right)\right] & \text{for } 2.25 \leq K_s^+ \leq 90 \\ \frac{1}{\kappa}\ln\left(1 + C_s K_s^+\right) & \text{for } K_s^+ > 90 \end{cases} \tag{15}$$

where $K_s^+$ is the non-dimensional roughness

$$K_s^+ = \frac{\rho K_s u^*}{\mu} \tag{16}$$

where $u^*$ is friction velocity. In Eq. (17), the roughness height was assumed to be 0.01 m. Figures 9(a) and (b) show the effects of a rough surface on the torque and blade noise, respectively. Increasing the roughness decreased the torque by about 50% from 209 to 100. However, the noise on the blade surface increases significantly by about 15 db. Figure 9(c) shows that the maximum acoustic power of the turbulent flow decreased slightly. These results show that increasing the roughness reduced the boundary layer and vortex, but the main noise source became the oscillating interaction between the blade surface and the air rather than the oscillation of the turbulence and vortex.

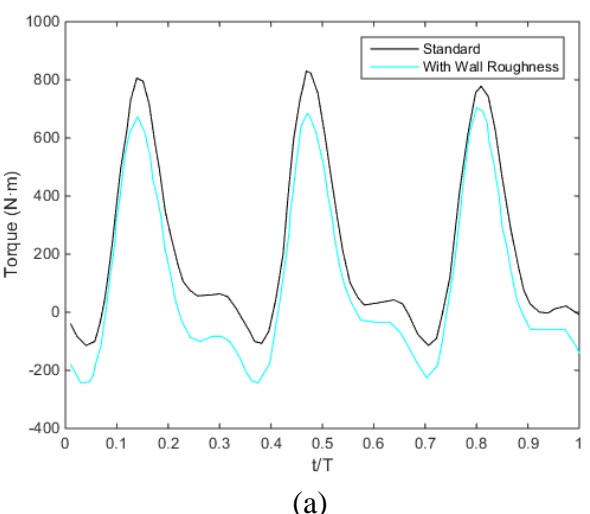

(a)

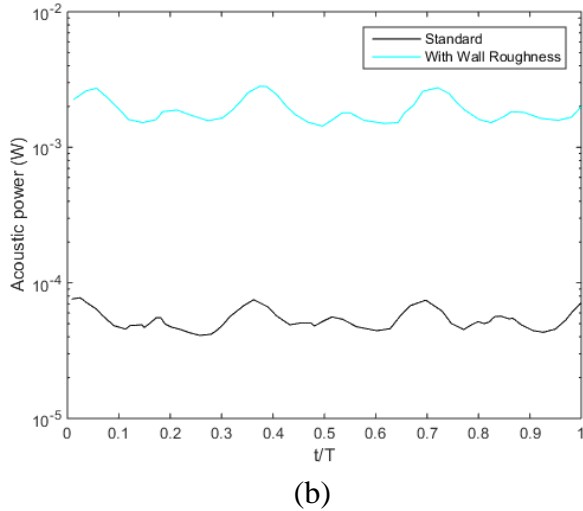

(b)

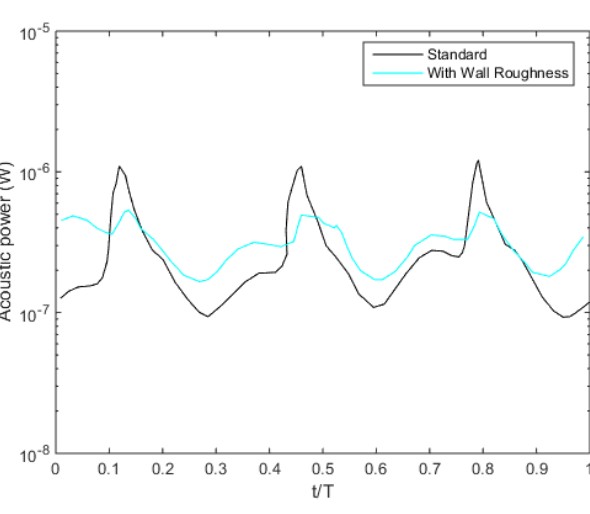

(c)

**Figure 9: Effects of the wall roughness on the (a) torque, (b) blade surface noise, and (c) turbulent flow noise.**

## 4. Conclusion

With the rapid advancement of computer science and technology, the use of CFD technology has become increasingly prevalent as a formidable means of exploring the aerodynamics of wind turbines. However, improving the performance and reducing the noise of VAWTs remains a difficult task due to their complex aerodynamic characteristics. ANSYS FLUENT is a powerful tool for investigating the aerodynamic and aeroacoustic behaviors of wind turbines, offering advantages such as low cost and provides a good flow
visualization. In this study, CFD simulations were performed to evaluate the effects of different noise reduction techniques on a small VAWT suitable for urban settings. A suitable mesh size and a turbulence model were determined, and the VAWT was initially evaluated to identify the reasons for torque reduction and noise increase. The choice of turbulence model is related to the operating condition and tolerance for y+ of the wind turbine. The SST k–ω turbulence model can better predict the flow field characteristics around
the wind turbine with or without noise reduction techniques as has been previously shown in the literature

for VAWTs. Then, three different noise reduction techniques were tested, and adding a deflector was found to increase the overall torque and decrease the blade noise. It also increased the turbulence noise, but this was still about 90% lower than the original blade noise (57.6 db). After installing the deflector, the torque of the blades increases by 169%, the acoustic power of the blades decreases by 98%, and the turbulent acoustic power increases up to 964%. The results of this study can serve as an important reference for other wind turbines under similar operating conditions and may contribute to the wider spread of small VAWTs in an urban setting. The aerodynamic performance enhancement of the small VAWT using the deflector can be done as future work, and it can be installed on the passage between the buildings and along the rooftop with a Fluid-Structure-Interaction study. This study is limited to only a numerical study using CFD techniques and steady-state simulations. Any airflow unsteadiness and variations in the intermittency and variability of wind speed and rotational speed were not considered in detail. There is only one tip speed ratio scenario examined in this study. The results are required further simulations and experiments to be an accurate study. Furthermore, URANS models will limit the size of the vortices to the very large ones, resulting in sound emissions that would probably be in the infrasound range. URANS models are also more suitable for predicting low frequency fluctuations and large-scale flow structures. The audible frequency range covers a broad spectrum, including both low and high frequencies, and URANS models may not easy to capture the entire range effectively. To investigate how acoustic energy is distributed over frequencies, we can perform a frequency analysis on the acoustic signal. This process involves transforming a time-domain signal into the frequency domain, revealing the different frequency components in the signal. The study that noise emissions focusing on different parameters in the infrasound range is the future study. In the meantime, the impacts of uniform and non-uniform building arrangements in an urban area are not yet taken into consideration. Higher-order models with unsteady wind conditions, tubercle amplitude-wavelength optimization, experimental analysis, incorporating end plates and supporting structures, and design optimization of the wind turbine parameters will also be the future studies.

## Competing interests

The contact author has declared that none of the authors has any competing interests.

## Acknowledgments

Support from the Atomic Energy Council (AEC), Taiwan is acknowledged.

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
