# Peer review of "Numerical model for noise reduction of small vertical-axis wind turbines"

_Wind Energy Science, 2023_

## Referee Comment (RC1)

**Review of article titled "Numerical model for noise reduction of small vertical-axis wind turbines (WES-2023-54)"**

by Kartik Venkatraman, von Karman Institute for Fluid Dynamics, Belgium

The topic addressed by the authors is relevant to the journal in terms of wind turbines and noise reduction strategies, but the work and the methodology has several strong short comings and lack of information which prevent me from accepting this manuscript in its current form and requires several major revisions. Please find attached the main general comments of concern, followed by specific comments.

**Main general comments**

1. The grid independence study has not been properly conducted, by increasing / decreasing the grid size over the entire domain, but parts of the domain have been selectively modified to justify grid independence (Table 1). The size of the mesh near wall region plays a key role in the predictions and this has not been changed (kept constant at 0.003 m) between Mesh 3 and Mesh 4, and the changes made in the mesh sizes is moreover negligible. Also importantly, a convergence in time (the number of revolutions, the time taken for the initial transient and convergence in forces) and the convergence with choice of time-step has not been shown. The mesh sizes could be reported based on number of points across the blade and the maximum surface $y^+$ over the revolution of the blade. Hence the effects of the mesh has not been addressed in a consistent manner.

2. The methodology (Section 2) is incomplete, with several variables and constants undefined, which makes it difficult to understand the model and choice. The constants used for the different turbulence models is missing. Several figures and numbers in the text have no units defined, and a consistent comparison has not been made eg. in some places the noise change is reported in dB, while others only a change in acoustic power (in W) is reported. And the methodology to compute the torque, angle of attack etc are not described.

3. No validation of either torque and/or noise measurements has been performed with any experimental/field measurement or other numerical data, which makes it difficult to have confidence in the simulation model. This along with the lack of a proper grid independence study makes the comparison for using the model for a study on different noise reduction methods (mask, deflectors, wall roughness) not reliable.

4. The CFD flow solution has been predicted using an in-compressible code (no variation of density and propagation of sound waves in the domain), relevant for only prediction of forces over the blade and a tonal noise prediction. The aeroacoustic formulation used for the study does not provide sound sources in the frequency spectrum and consider the Doppler shift that takes place over the revolution of the turbine. In this context, the entire discussion of noise and the acoustic power using turbulent kinetic energy does not seem appropriate as the acoustics are not captured by the simulated CFD model. Ideally an FW-H approach could to be performed using a converged blade loading statistics to estimate the tonal components, or switching to a compressible flow solver/methodology to capture the entire acoustic spectrum.

And an entire acoustic spectrum would be required to identify the different noise sources and their differences for then simulating different noise reduction methods.

5. The Introduction (Section 1) does not appear to be thorough and complete about the most recent literature related to noise radiated by vertical axis wind turbines, and contains too many references to work not directly related to the present study (vertical axis wind turbine noise and noise reduction strategies). Moreover the literature presented does not highlight the methodology and important simulation details to draw a contrast and highlight the novelty of the present work.

6. The conclusion about the $k - \omega$ SST turbulence model showing a better prediction for the flow field is not novel (Line 359), as it has been used significantly in published literature for vertical axis turbines. Also the noise reduction has been reported in percentages but not in dB in conclusions. Moreover, the deflector suitable noise reduction has not been been shown for all operating conditions of the turbine (only a single velocity ratio has been simulated), hence it is not a conclusive (Line 365) justification to use it as a noise reduction approach.

**Specific comments**

1. Line 32: 95 dB sound pressure level (SPL). Which kind of turbine (horizontal axis or vertical axis) does it correspond to ?

2. Line 35: Rephrase, "noise by air"

3. Line 39: "acoustic analogy theory" : This statement does not seem completely correct. There is the direct noise prediction (by performing a direct numerical simulation or large eddy simulation) or indirect noise approach by separating the noise source calculation, and noise propagation. And the acoustic analogy is one of the ways for noise propagation (eg. other approaches such as finite element, boundary element, or low order approaches).

4. Line 40: Inconsistent use of sound pressure / sound pressure level (SPL), I suggest to define and use the abbreviation

5. Line 43: ANSYS CFX version ? Which method (RANS or LES) ?

6. Line 44: Relevance of NREL Phase VI blade ? Is it a VAWT ?

7. Line 49: Tip noise is the dominant source ? Is it true for VAWTs ? Justification/citation to relevant literature required

8. Line 51: Maizi.et al (2018) used a 2D or 3D approach ?

9. Line 57: The term "velocity ratio' is not defined

10. Line 61: What is meant by spacing ? Does it refer to blade solidity ? Any justification given in the paper for "excessively small or large spacing increasing noise emissions ?

11. Line 62: The review of Botha (2017) is vague regarding the analytical model falling short ? Which noise generation mechanism was modelled ?

12. Line 69: Naccache et. al (2018) , appears to be irrelevant to the present study, since it is a D-VAWT and not a standard VAWT that has been investigated in the present study.

13. Line 77: Ideally always method the methodology used in a flow solver (RANS/LES/DNS etc) and if the simulation is 2D or 3D.

14. Line 79: Nyborg et. al (2018) the use of higher -fidelity sound propagation model is vague. Is it a empirical model ? Is it relevant to the present study on VAWTs ?

15. Line 92: Not true, there have been several studies which are missing from the literature review such as for example: https://www.mdpi.com/1996-1073/13/16/4148, https://journals.sagepub.com/doi/10.1260/1475-472X.14.5-6.883, https://arc.aiaa.org/doi/10.2514/6.2022-3058

16. Line 102: ANSYS FLUENT has to be mentioned unsteady RANS (uRANS) approach is being used, incompressible flow can capture only tonal components (unsteady blade loading). For turbulence interaction noise, no propagation of sound waves since the density is constant. , the numerical schemes and methodology is too dissipative.

17. Line 116: The relevant research related to the 2 turbulence models that have been extensively used for VAWTs has not been cited.

18. Line 120: The constants have not been defined. Please define all the constants and the values for the constants in a Table for both the turbulence models.

19. Line 138: $\epsilon$ has not been defined

20. Line 145: Relevance of the used methodology/formulation for rotating machines ? Has it been used in published literature for rotating machines and also specifically for VAWTs ? The variation of noise sources over the revolution of the blade is not accounted for.

21. Line 151: What is the correlation area ? How is it defined ? A diagram could be useful.

22. Line 157: No end plates or supporting structures have been included in the model. They could alter in the predicted blade loading and noise characteristics, in case a practical noise reduction methodologies are required. https://doi.org/10.1108/HFF-09-2022-0562

23. Line 183: How accurate are first-order schemes ? The turbulent kinetic energy could be too dissipative to have any meaningful prediction for the computed noise (the methodology which itself is questionable for use for VAWT noise prediction).

24. Line 184: Influence of time step size has not been reported ? How is this time step choice justified ? Is it based on a CFL number / any relevant flow physics ?

25. Line 210: Refer main general comment 1, also the number of points across the blade and the maximum wall $y^+$ could be reported in the table. The mesh sizes have not been changed in a consistent way.

26. Line 221: The methodology used to compute the torque is not reported. Are the forces on the blades summed up, and multiplied with radius and the rotational speed ?

27. Line 224: Mesh independence study has not been performed in consistent way, by changing the mesh size over the entire domain , refer main general comment 1.

28. Line 230: Time-averaged torque of 227.7 , unit undefined.

29. Line 240: Figure 8 : unit undefined for Torque , unit for acoustic power shown in W, but the differences reported in dB in text.

30. Line 245: Figure 3, how many revolutions of the turbine have been simulated ? Has convergence been achieved in time. How long does it take for the initial transient ?

31. Line 256: How were the angles of attack calculated ? Was the flow velocity sampled at a point ?

32. Line 296: Deflectors could be similar to trailing edge serrations used for noise reductions. Agree that they help energize vortices close to the trailing edge and reduce extent of separation.

33. Line 320: No units on legends for Pressure, also the azimuthal angle (positions of the blades over the revolution) needs to be added.

34. Line 336: $u*$ is not defined

35. Line 351: The higher wall roughness could increase the wall pressure fluctuations which increase the noise. But here the increase/change has not been reported in dB.

36. Line 355: "visualization effect of the distribution" - unclear, please rephrase.

37. Line 358: Refer main general comment 6

---

## Author Comment (AC1)

Reply to the Reviewer #1

The topic addressed by the authors is relevant to the journal in terms of wind turbines and noise reduction strategies, but the work and the methodology has several strong short comings and lack of information which prevent me from accepting this manuscript in its current form and requires several major revisions. Please find attached the main general comments of concern, followed by specific comments.

Reply: The authors would like to thank the reviewer for the time to review our paper. The comments that the reviewer provided have contributed to the enhancement of our paper. We have taken the opportunity to make several improvements in the text, in order to strengthen the paper. A list of point-by-point replies to the reviewer's comments is reported in the following.

Specific comments

1. Line 32: 95 dB sound pressure level (SPL). Which kind of turbine (horizontal axis or vertical axis) does it correspond to?
   Reply: HAWT can produce about 95 sound pressure level (dB) of noise. But we deleted the content since HAWT is irrelevant to the present study.

2. Line 35: Rephrase, "noise by air"
   Reply: We changed the sentence based on reviewer's suggestion.

3. Line 39: "acoustic analogy theory" : This statement does not seem completely correct. There is the direct noise prediction (by performing a direct numerical simulation or large eddy simulation) or indirect noise approach by separating the noise source calculation, and noise propagation. And the acoustic analogy is one of the ways for noise propagation (eg. other approaches such as finite element, boundary element, or low order approaches).
   Reply: Thanks for pointing out. We have deleted the content.

4. Line 40: Inconsistent use of sound pressure / sound pressure level (SPL), I suggest to define and use the abbreviation
   Reply: We defined the abbreviation of SPL.

5. Line 43: ANSYS CFX version ? Which method (RANS or LES) ?
   Reply: We deleted the content since NREL Phase VI HAWT is irrelevant to the present study.

6. Line 44: Relevance of NREL Phase VI blade ? Is it a VAWT ?
   Reply: We deleted the content since NREL Phase VI HAWT blade is irrelevant to the present study.

7. Line 49: Tip noise is the dominant source ? Is it true for VAWTs ?
   Justification/citation to relevant literature required
   Reply: We deleted the content in the paper.

8. Line 51: Maizi.et al (2018) used a 2D or 3D approach ?
   Reply: They conducted a 3D numerical analysis with unsteady CFD simulations. We have added the text in the paper.

9. Line 57: The term "velocity ratio' is not defined
   Reply: We changed the sentence velocity ratio into tip speed ratio.

10. Line 61: What is meant by spacing ? Does it refer to blade solidity ? Any justification given in the paper for "excessively small or large spacing increasing noise emissions ?
    Reply: Spacing meant between the airfoils in every blade. We added in the discussion. We also deleted the sentence excessively small or large spacing increasing noise emissions.

11. Line 62: The review of Botha (2017) is vague regarding the analytical model falling short ? Which noise generation mechanism was modelled ?
    Reply: We have extensively revised the content of this literature review.

12. Line 69: Naccache et. al (2018) , appears to be irrelevant to the present study, since it is a D-VAWT and not a standard VAWT that has been investigated in the present study.
    Reply: We deleted the content since Dual VAWT is irrelevant to the present study.

13. Line 77: Ideally always method the methodology used in a flow solver (RANS/LES/DNS etc) and if the simulation is 2D or 3D.
    Reply: We added more information in the paper review.

14. Line 79: Nyborg et. al (2018) the use of higher -fidelity sound propagation model is vague. Is it a empirical model ? Is it relevant to the present study on VAWTs ?
    Reply: We deleted the content since this paper review is irrelevant to the present

study.

15. Line 92: Not true, there have been several studies which are missing from the literature review such as for example: https://www.mdpi.com/1996-1073/13/16/4148, https://journals.sagepub.com/doi/10.1260/1475-472X.14.5-6.883, https://arc.aiaa.org/doi/10.2514/6.2022-3058
Reply: We added the three papers in the literature review based on reviewer's suggestion.

16. Line 102: ANSYS FLUENT has to be mentioned unsteady RANS (uRANS) approach is being used, incompressible flow can capture only tonal components (unsteady blade loading). For turbulence interaction noise, no propagation of sound waves since the density is constant, the numerical schemes and methodology is too dissipative.
Reply: We deleted the dissipative text based on reviewer's suggestion.

17. Line 116: The relevant research related to the 2 turbulence models that have been extensively used for VAWTs has not been cited.
Reply: The relevant research (Venkatraman et al. (2021); Mohamed (2016)) related to the 2 turbulence models have been added in the paper.

18. Line 120: The constants have not been defined. Please define all the constants and the values for the constants in a Table for both the turbulence models.
Reply: We define all the constants and the values for the constants in Table.1 based on reviewer's suggestion.

19. Line 138: $\epsilon$ has not been defined
Reply: $\epsilon$ has been defined.

20. Line 145: Relevance of the used methodology/formulation for rotating machines ? Has it been used in published literature for rotating machines and also specifically for VAWTs ? The variation of noise sources over the revolution of the blade is not accounted for.
Reply: Dinulovic et al. presents the aeroacoustic calculation methodology for the

H-Darrieus wind turbine. The CFD analysis, for different wind turbine blades' angles of attack, coupled with the noise analysis is calculated. In this paper, sound propagation equation is solved by Lighthill and Curle. We added the text in the paper.

(Dinulovic M, Trninic M, Rasuo B and Kozovic D (2023) Methodology for aeroacoustic noise analysis of 3-bladed h-Darrieus wind turbine, Thermal Science, 10.2298/TSCI2301061D, 27:1 Part A, (61-69).)

Nukala et al. described that FW-H and Curle's analogies are the most widely used integral methods for predicting acoustic field and considered as formal solutions of Lighthill's equation applied for rigid and moving wall boundaries relative to flow field.

(Vasishta Bhargava Nukala & Chinmaya Prasad Padhy (2023) Concise review: aerodynamic noise prediction methods and mechanisms for wind turbines, International Journal of Sustainable Energy, 42:1, 128-151, DOI: 10.1080/14786451.2023.2168000)

21. Line 151: What is the correlation area ? How is it defined ? A diagram could be useful.

    Reply: Based on reviewer's comment, we have defined the correlation area.

22. Line 157: No end plates or supporting structures have been included in the model. They could alter in the predicted blade loading and noise characteristics, in case a practical noise reduction methodologies are required. https://doi.org/10.1108/HFF-09-2022-0562

    Reply: In the past, many studies only considered the wind turbine blades when simulating blade noise. This method not only allowed for a more accurate assessment of the blades' torque and noise, but also contributed to understanding the aerodynamic characteristics. However, the reviewer's comment is also reasonable. Comprehensive simulation of the entire wind turbine noise characteristics and the turbine performance, considering the end plates and supporting structures might be necessary. This could be incorporated into the future works.

    (Viqueira-Moreira M, Ferrer E. Insights into the Aeroacoustic Noise Generation for Vertical Axis Turbines in Close Proximity. Energies. 2020; 13(16):4148. https://doi.org/10.3390/en13164148)

23. Line 183: How accurate are first-order schemes ? The turbulent kinetic energy could be too dissipative to have any meaningful prediction for the computed noise (the methodology which itself is questionable for use for VAWT noise prediction). Reply: The equations coupling velocity and pressure are derived from the continuity equation, basically the continuity equation is not easy to converge, so the 2nd order scheme is adopted. However, the turbulent convergence of the cases in this paper, there aren't significant problem with turbulent flow convergence, so 1st order can be used.

24. Line 184: Influence of time step size has not been reported ? How is this time step choice justified ? Is it based on a CFL number / any relevant flow physics ? Reply: In this study, an implicit scheme is employed to calculate the transient terms, so the acoustic cases were less sensitive to the CFL number. CFL number is acceptable around 1 based on the FLUENT manual. Therefore, the CFL number 0.12 to 2 and a time step of 0.01 s were used in the paper. We added this information in section 2.5 in the revised paper.

25. Line 210: Refer main general comment 1, also the number of points across the blade and the maximum wall y+ could be reported in the table. The mesh sizes yhave not been changed in a consistent way. Reply: SST k–ω model has a higher tolerance for y+, therefore y+ has no impact on the results. To ensure the computational validity of realizable k–ε, maximum y+ value less than 80 in this paper. We added the information in section 2.4.

26. Line 221: The methodology used to compute the torque is not reported. Are the forces on the blades summed up, and multiplied with radius and the rotational speed ? Reply: We added the methodology used to compute the torque in the paper.

27. Line 224: Mesh independence study has not been performed in consistent way, by changing the mesh size over the entire domain, refer main general comment 1. Reply:
Based on the recommendations in the ASME V&V 20-2009 report regarding establishing the mesh independence study, ASME suggests changing the mesh size by approximately 1.3 times based on their experience to ensure a significant difference. In this study, we also performed the mesh independence study based on this report.

28. Line 230: Time-averaged torque of 227.7 , unit undefined.
    Reply: We added the units in the sentence.

29. Line 240: Figure 8 : unit undefined for Torque , unit for acoustic power shown in
    W, but the differences reported in dB in text.
    Reply: We added the units in the figure, and changed db to W in the text.

30. Line 245: Figure 3, how many revolutions of the turbine have been simulated ?
    Has convergence been achieved in time. How long does it take for the initial
    transient ?
    Reply:
    60 rpm
    Yes
    About 1 min

31. Line 256: How were the angles of attack calculated ? Was the flow velocity
    sampled at a point ?
    Reply: The attack angle is the angle at which the chord of a blade meets the wind
    velocity. The wind velocity was sampled at a point. We added this information in
    the revised paper.

32. Line 296: Deflectors could be similar to trailing edge serrations used for noise
    reductions. Agree that they help energize vortices close to the trailing edge and
    reduce extent of separation.
    Reply: Thanks for the reviewer's agreement.

33. Line 320: No units on legends for Pressure, also the azimuthal angle (positions of
    the blades over the revolution) needs to be added.
    Reply: We added the units on legends in Fig. 4 and 8 in the revised paper.
    Different t/T values correspond to blade positions. For instance, t/T=0.5 means a
    rotation of 180 degrees. We also added this information in section 3.1.

34. Line 336: u∗ is not defined
    Reply: We defined it in the sentence.

35. Line 351: The higher wall roughness could increase the wall pressure fluctuations

which increase the noise. But here the increase/change has not been reported in dB.

Reply: We added the noise 15 db in the text.

36. Line 355: "visualization effect of the distribution" - unclear, please rephrase.

    Reply: We changed the phrase in the sentence.

37. Line 358: Refer main general comment 6

    Reply: The authors made several modifications in the conclusion based on reviewer's comment.

Thanks for the valuable comments.

---

## Author Comment (AC2)

Reply to the Reviewer #3

This manuscript presents an investigation of aerodynamic noise emission of a small vertical-axis wind turbine in different configurations designed to alter and reduce noise emissions.

For the most part, the article is clearly written. However, the study of aeroacoustics is conducted with tools that are not well suited for the refined analysis required for estimating noise emission reductions in details and consequent environmental impact. The paper also suffers a number of deficiencies that are addressed following the sections' order below.

The introduction starts with generalities about wind turbine noise. The literature review appears a bit random and disorganized.
Reply: We have reorganized the introduction and paper review in the revised paper based on reviewer's comment.

In Section 2, the numerical models are presented. The flow solution is based on a commercial software solving the compressible Navier-Stokes equations, with a standard RANS approach for turbulence modelling. In this respect, the unsteady term is missing in Eq.(2). The acoustic part is based on an integral acoustic analogy derived from Curle analogy. The integration surfaces for the acoustic calculations are on the blade surfaces. Note here, that strictly speaking, Curle's analogy is only valid for non-moving surfaces. To the authors' credit, in the present case, it may be a valid approximation though. A "correlation area" is introduced as a multiplicative factor in Eq.(11), but it is never numerically defined. Since this factor is related to wall-pressure correlation lengths, its value has a direct and significant impact on the noise emission levels. The accuracy of the overall model can then be questioned if it is not properly defined based on some physical considerations. In addition, the overall methodology for acoustic predictions seriously lacks qualitative validations, or at least references to it.
Reply: We changed the eq (2) and defined the $A_c$ in eq (13) in the revised paper based on reviewer's comment. We also defined the overall model on some physical considerations. We added some references in section 1 and 2 based on reviewer's comment.

Furthermore, as shown in the results, the model provides only an overall (frequency-integrated) sound power level. There is no indication on how this energy is distributed

over frequencies, which is a crucial aspect for environmental impact of wind turbines (or any other industrial facility for that matter). Not even an indication on A-weighted noise power levels can be obtained.

Reply: The main objective of this paper is to address how to reduce the magnitude of noise, focusing primarily on the overall (frequency-integrated) sound power level, which is also a concern for wind turbine operators. In the next stage, our research team will undertake a related study and noise analysis focusing on parameters such as noise energy vs frequency. Thanks for the reviewer's suggestions.

To make things worse, the model as it is, can most probably not capture scattering effects, which should be the dominant noise source in this context (at least in the audible frequency range). It may be that the present model can capture vortex-blade interaction noise. However, the use of a URANS strategy will limit the size of the vortices to the very large ones, resulting in sound emissions that would probably be in the infrasound range. It can be expected that this has little relevance in the present context.

Reply: URAN models may underestimate the size of vortices, potentially leading to an underestimation of noise in the infrasound range. LES model may provide a more accurate understanding of these phenomena. However, LES model requires significant computational resources. For industrial applications, the SST k-$\omega$ model currently offers sufficient accuracy with an appropriate computational load, making it a good choice. Additionally, this study primarily focuses on the modification of wind turbine blades to reduce noise. Even if the noise in the infrasound range cannot be accurately estimated, as shown in Figures 6-9, the SST k-$\omega$ model can still be used to identify methods for noise reduction, making it applicable for industrial assessments.

In Section 2.4, the numerical implementation and discretization of the CFD model is presented. It is classical for these type of calculations to provide the size of the cells at the surface (for assessing mesh refinement) in term of y+ (as mentioned by the author in the introduction). But, this is never done here. Only cell sizes in meter are provided in Table 1.

Reply: SST k−ω model has a higher tolerance for y+, therefore y+ has no impact on the results. To ensure the computational validity of realizable k–ε, maximum y+ value less than 80 in this paper. We added the information in section 2.4 based on reviewer's comment.

Section 3 starts with a mesh convergence analysis which appears satisfying. The

remaining of the investigation concentrates on noise emissions which is lacking a number of information to make it valuable. The comparisons concentrates on the impact of the geometry for noise reduction on torque and acoustic power. This is indeed a suitable approach as indeed, acoustic reduction should affect overall aerodynamic performances to a minimum, which should be addressed in the design phase. Nevertheless, the lack of physics in the noise modelling (see earlier comment about Section 2) does not permit to draw firm conclusions on the impact of the different designs, as the noise reduction (or increase) may occur at frequencies that are not relevant for environmental purposes.

Reply: Reviewer is correct. This paper acknowledges that the noise reduction design may not be applicable to conditions that are not relevant for environmental purposes. However, the main objective of this paper is to improve the design of wind turbine to reduce the impact of turbine blade rotation on environmental noise. Therefore, the conclusions of this paper can still be considered as an important reference.

As an additional comment, the authors use a very unusual metric for displaying the noise levels, namely watts. Note values up to $10^5$ W in Fig.3 which is of the order of the noise emission of a turbojet engine....

Reply: The values are around $10^{-4}$ W in Fig 3. in the revised paper.

To conclude, the paper presents an analysis of VAWT noise emissions which is questionable regarding its relevance for assessing environmental impact. Nonetheless, if interested in very qualitative acoustic results, that may be satisfying, e.g. if one is mainly interested in aerodynamic performances of the acoustic reduction devices. In the reviewer's opinion, the acoustic model strategy should be improved/enhanced, and noise spectra should be provided as a result, before this article can be considered for publication.

Reply: This paper primarily employs CFD method to investigate the enhancement of VAWT design with the purpose of reducing noise. Therefore, the focus is on the overall (frequency-integrated) sound power level, which is of particular concern to wind turbine operators. As for the impact of improving blade design on aerodynamic performances, an investigation is required into its relationship with the energy vs frequency of noise (rather than only calculating the overall power level). This is the next research topic for our team.

Thanks for the valuable comments.

---

## Author Comment (AC3)

Reply to the Reviewer #2

The paper addresses an interesting topic but I feel like it misses a validation of the baseline simulation first. This should be done before proceeding to testing mitigations in the simulation.

comments:

-line 175: why is the outer volume a cylinder? I had expected a rectangle with a wall boundary condition to represent the ground/building on which the turbine is installed.
Reply: Cylindrical domains are often employed for simulating individual wind turbine blades or small wind turbine systems. The geometric similarity between cylinders and wind turbine blades allows for a better representation of the airflow around the blades. However, rectangular domains are more suitable for simulating larger wind fields, such as interactions between multiple turbines or airflow distributions within a wind farm. The rectangular shape aids in comprehensively capturing the overall characteristics of the wind field but might lack the detailed flow patterns of individual blades.

- why is dynamic mesh used and not moving mesh? Is the mesh quality around the airfoil good during all time steps? Can you please include plots showing details of the mesh around the airfoil? In case moving mesh is used, one can control better the quality of the mesh around the airfoil.
Reply: In this study, the rotating zone refers to the dynamic mesh of rotation, without using moving mesh, and the rotating frequency is 60 rpm. The mesh independence test can demonstrate the mesh quality, and we show details of the mesh around the airfoil based on reviewer's comment. The mesh quality around the airfoil is good during all time steps. Dynamic mesh allows for adaptability to complex geometries without requiring re-meshing. An implicit scheme is employed to calculate the transient terms, so the acoustic cases were less sensitive to the CFL number. CFL number is acceptable around 1 based on the FLUENT manual. Therefore, the CFL number 0.12 to 2 and a time step of 0.01 s were used in the paper. We added this information in section 2.5 in the revised paper.

- what is the courant number you have with the selected time step?
Reply:
CFL=u*dt/dx, u=12 m/s, dt = 0.01 s, dx=0.06 to 1.
So the CFL=0.12 to 2 in this paper.

- the near-wall region and outer-flow region were not refined/made coarser to confirm mesh convergence. I think this is a necessary step to be sure about the convergence. Also a convergence study on the time step is necessary.

Reply: Mesh independence calculations have been performed and shown in Fig. 3 (a) and (b), and Table 2 in the revised paper.

-the acoustic power plot doesn't have 3 similar periods. Do you have an explanation for that?

Reply: Because the insufficient mesh numbers in Mesh 1 and 2, they might lead to less accurate results. Hence, the acoustic power plot doesn't have 3 similar periods. Similar results were obtained for the predicted acoustic power on the blade, as shown in Fig. 3(b). Mesh 3 and Mesh 4 resulted in nearly identical predictions, so Mesh 3 was selected for subsequent analysis because it had a smaller grid number and thus was less computationally intensive.

small comments:

- 1st sentence of the abstract: "Wind turbines are a promising solution for sustainable energy": I suggest to narrow it down to Small vertical-axis wind turbines. The noise reduction methods for horizontal wind turbines are different than what is described in the following sentences.

Reply: Thanks for pointing out. We have changed the content.

- line 33: these numbers don't match the typical formula used in IEC61400-11.

Reply: Based on reviewer's comment, we have deleted the content.

Thanks for the valuable comments.

---

## Author Response (AR2)

Reply to the Associate editor:

The authors have addressed most of the reviewer's specific comments but I believe more could be done in the following areas:

Scope of the work: It is important right from the beginning that the authors make clear that this work is more of a qualitative study looking at changing VAWT design to improve performance and reduce noise and not an accurate study of the environmental noise impact of a VAWT.

Limitations of the model and the simulations: The authors need to state the limitations of the model and note that only one tip speed ratio scenario has been studied. Thus, it is difficult to draw firm conclusions from the results and further simulations are required.

Future work: I believe the authors could expand on this area.

To this end I have made some edits and comments in the attached file.

Reply: We have made comprehensive revisions in the revised paper based on the associate editor's suggestions, including Scope of the work, Limitations of the model and the simulations, Future work, and some edits and comments in the attached file. We deeply appreciate the assistance and valuable suggestions.